# Single-cell measurement quality in bits

**Jayan Rammohan, Swarnavo Sarkar, David Ross**[ORCID] *

National Institute of Standards and Technology, Gaithersburg, MD, United States of America

* david.ross@nist.gov

## Abstract

Single-cell measurements have revolutionized our understanding of heterogeneity in cellular response. However, there is no universally comparable way to assess single-cell measurement quality. Here, we show how information theory can be used to assess and compare single-cell measurement quality in bits, which provides a universally comparable metric for information content. We anticipate that the experimental and theoretical approaches we show here will generally enable comparisons of quality between any single-cell measurement methods.

**Data Availability Statement:** All experimental data are publicly-available through the NIST Data Portal (https://doi.org/10.18434/mds2-2300). Data analysis code is available at https://github.com/sarkar-s/MQuIT.

## Introduction

The development of single-cell measurements has revealed that cellular sense-and-response within a population of isogenic cells is noisy [1–3]. The interpretation of this biological noise has directly led to improvements in our ability to understand and engineer biological systems [4, 5]. Importantly, however, the measurement process *itself* includes noise. So, the results of single-cell measurements contain biological noise as well as measurement noise. Unfortunately, relatively few studies have been performed to understand and compare the quality of single-cell measurement methods, which could inform our interpretation of biological noise [6, 7] and selection of methods [8–10]. In general, single-cell measurement quality is not well-defined. Various statistical metrics have previously been used to evaluate single-cell measurement quality, such as signal-to-noise ratio (SNR) [4] and area under the receiver operating characteristic curve (AUC) [10], however the validity of these performance metrics relies on assumptions about the measured distributions. For example, SNR calculations implicitly assume that the underlying biological distributions are approximately Gaussian. Furthermore, these performance metrics use units which are neither intuitive nor universally comparable.

We propose that information theory can be used to understand and evaluate single-cell measurement quality in units that are both intuitive and universally comparable: bits. Key aspects of information theory were developed from efforts to understand signal processing and communication in the presence of noise [11–14], and there are useful textbooks on the foundations of information theory [15] as well as its application to biological systems [16]. A common unit of information is a binary digit, or "bit", which intuitively represents the ability to distinguish between two states. Therefore, an assessment of measurement quality in bits could provide intuition about how well different single-cell measurement methods can distinguish between different cell states. More specifically, if we consider the measurement process

**Funding:** The author(s) received no specific funding for this work.

**Competing interests:** The authors have declared that no competing interests exist.

as communication through a noisy information channel, the *mutual information* between the input and output of that channel quantifies how much information is shared, or transmitted, through the channel. Mutual information depends on both the communication channel and the probability distribution of possible input signals. The maximum mutual information for all possible input distributions is the *channel capacity*, which is a characteristic property of communication channels. Channel capacity (in bits) is the base-2 logarithm of the maximum number of distinguishable input signal levels. Intuitively, a channel capacity of one bit indicates that a measurement can distinguish between cells grown in two different levels of environmental stimulus.

Here, we show how an information-theoretic approach can be used to assess and compare the quality of single-cell measurement methods in bits. Using the channel capacity between an environmental stimulus and the measured response as a metric, we interpret and compare the quality of multiple methods for measuring RNA or protein in single cells. We find a wide range in the amount of information that different methods can transmit about single-cell gene expression. Furthermore, to show how an information theoretic analysis can inform our choices for steps of the measurement process, we show how changes to specific steps can impact measurement quality. This generalizable approach offers a way to assess and compare the measurement quality of different single-cell methods in universally comparable units.

## Results

To quantitatively assess and compare single-cell measurement quality using information theory, we considered the fundamental question: How well can a measurement estimate the biological response (output) to an environmental stimulus (input)? One common approach for studying cellular response is induction of gene expression, in which an environmental stimulus, such as the concentration of an inducer molecule, causes a change in the level of gene expression inside the cells. In this case, the question can be rephrased as: How well do single-cell measurements of gene expression transmit information about the way cells respond to an environmental stimulus?

It is challenging to compare the quality of different single-cell methods across different studies, because biological variability is confounded with experimental variability. For example, experimental variability introduced by different cell culture conditions can influence cellular function [17], therefore different single-cell methods performed under different conditions may not provide direct insight into differences in measurement quality. So, to minimize the influence of experimental variability on method comparison, we used a recently-reported collection of data from different single-cell measurements of the same underlying biological system: inducible gene expression in *E.*coli [10]. In that study, a split-sample approach was used to measure cells harvested from the same replicate cultures using different single-cell methods. For each biological replicate, cells were divided (split) at each step of the measurement process: once for sample preparation, again for signal detection, and finally for choice of measurand. In this manner, multiple single-cell methods were performed while minimizing the contributions of experimental variability. The methods included different measurands (RNA, fluorescent protein) and signal detection with different instruments (microscopy, flow cytometry). Also, different sample preparation methods were used for each measurand, including two different methods for specific, fluorescent labeling of an RNA transcript: fluorescence *in situ* hybridization (FISH) and hybridization chain reaction (HCR). For microscopy images of RNA, single-molecule localization was used to estimate RNA abundance per cell. Tetramethylrhodamine (TAMRA) was used for fluorescence labeling in the FISH and HCR methods, and yellow fluorescent protein (YFP) was used as the protein measurand. The fluorescence spectra of

TAMRA and YFP are distinct. So, after RNA labeling, both RNA and protein could be measured in the same set of cells using different channels on the flow cytometer or different filter sets on the microscope. For each method, cellular response was measured by detecting gene expression levels (RNA or fluorescent protein) from cells cultured over the entire range of inducible response. Here, we use the experimental results from [10] to show how channel capacity can be used as a metric to assess differences in single-cell measurement quality.

To realize a generalizable metric for the quality of different single-cell measurements, we used the empirical gene expression distributions at each inducer concentration along with the Blahut-Arimoto algorithm [18–20] to calculate the channel capacity between the environmental input and the measured gene expression output (Fig 1A). Details of the implementation of the Blahut-Arimoto algorithm are described in the Materials and Methods. Briefly, for each sample and each measurement, we started with the single-cell measurement results: a list of numbers corresponding to the measurement result for each cell in a sample. We binned those results and normalized the number of observations in each bin to give the discrete empirical distributions of the measurement results. These empirical distributions represent the *conditional* distributions for each fixed value of the environmental stimulus (i.e., each sample). They

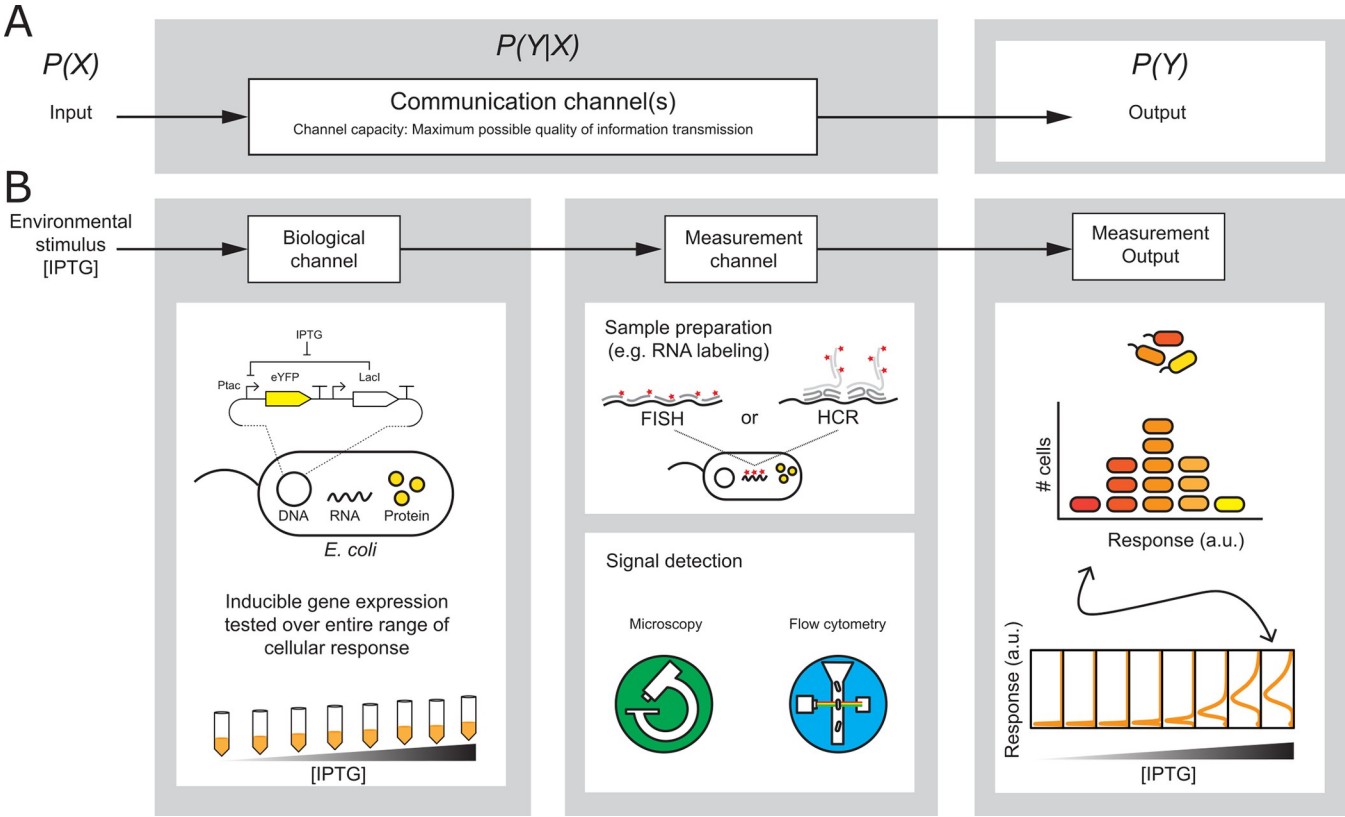

**Fig 1. Single-cell measurement quality in bits can estimated from the channel capacity between environmental stimulus and measurement output.** (A) An information theoretic "communication channel" transmits information from an input signal with probability distribution *P(X)* to an output signal *P(Y)* in the presence of noise. The "channel capacity" is a characteristic of the channel representing the maximum possible quality of information transmission. (B) Single-cell measurements of cellular response to environmental stimulus can be interpreted by considering the biological channel and measurement channel in series. The channel capacity of the input (environmental stimulus) to measurement output (estimated levels of single-cell gene expression over the range of response) characterizes information transmission through the biological channel and the measurement channel. Single-cell methods can vary with regards to sample preparation, for example RNA labeling by fluorescence in situ hybridiation (FISH) or hybridization chain reaction (HCR). Single-cell methods can also vary with regards to signal detection, for example, microscopy or flow cytometry. Calculations of channel capacity using different measurement channels for the same biological channel enables an assessment of single-cell measurement quality in bits.

are used by the Blahut-Arimoto algorithm, which numerically solves for the maximum mutual information (i.e., channel capacity) while varying the probability distribution of the input environmental stimulus.

Notably, the channel capacity is not determined purely by the quality of the single-cell measurement method; it also depends on the cellular response. So, to use channel capacity as metric for measurement quality, we considered a two-channel model with a *biological channel* and *measurement channel* connected in series (Fig 1B). In this model, the biological channel is the cellular response that transmits information from the environmental stimulus to changes in gene expression, and the measurement channel is the entire measurement process that transmits information from the *actual* gene expression to the *estimated* gene expression, including all steps such as sample preparation, signal detection, choice of measurand, and data analysis.

Since no measurement is perfect, the measurement channel will degrade the information that it transmits. So, the *measured channel capacity*, i.e. the channel capacity between the input and the estimated gene expression, will always be less than the *biological channel capacity*, i.e., the channel capacity between the input and the actual gene expression. Higher quality measurements, however, will degrade the information less. So, we can assess relative measurement quality by comparing the measured channel capacities for different measurement methods: Higher quality measurements will result in a higher measured channel capacity (i.e., closer to the true biological channel capacity). It is important to note, however, that this assessment of measurement quality requires the assumption that the measurement methods to be compared all share the same biological channel. With the split-sample datasets, we can justify that assumption, but only for comparisons between different methods with the same measurand. The datasets include measurements of two different measurands, RNA and fluorescent protein, that correspond to two different biological channels. So, in the assessment of measurement quality, we only compare channel capacity between measurements of the same measurand (i.e., we compared RNA methods only to other RNA methods, and protein methods only to other protein methods).

As a final consideration, it is important to note that the channel capacity can also depend on the choice of values used for the input stimuli. If the channel capacity is the logarithm of the number of distinguishable input levels, then it clearly cannot be greater than the logarithm of the number input levels measured. For example, if an experiment only uses two input levels (e.g., test and control, or high and low), then the channel capacity determined by our approach will always be less than or equal to one ($\log_2(2) = 1$). Furthermore, to obtain the best estimates of the biological channel capacity and the best comparison of different methods, the input levels should be chosen to span the full range of biological response. For example, the datasets used here include IPTG concentrations across the full induction curve, with input levels that result in low biological output (i.e., gene expression), high biological output, *and* intermediate biological output. In general, a different choice of input levels measured in an experiment could lead to different estimates of the channel capacity with our approach. So, for assessment of measurement quality, we recommend comparing only methods implemented with the same set of input levels (as with the split-sample dataset used here).

To evaluate single-cell RNA measurement quality in bits, we compared the channel capacities for different methods of measuring the same RNA expression system. For different single-cell RNA measurement methods, we observed a wide range of channel capacities (0.09 bits to 1.06 bits; Tables 1 and 2), which we attribute to differences in the quality of the measurement methods (Fig 2A). Flow cytometry detection of HCR-labeled RNA had the lowest channel capacity ($\approx 0.09$ bits). Microscopy detection of FISH-labeled RNA had the highest channel capacity ($\approx 1.06$ bits).

Table 1. Channel capacities of single-cell RNA measurements.

| Sample Preparation | Signal Detection | Channel capacity from environmental input to measurement output* |
|---|---|---|
| RNA labeling (FISH) | Microscopy | 1.06 ± 0.08 bits |
| RNA labeling (HCR) | Microscopy | 0.88 ± 0.22 bits |
| RNA labeling (FISH) | Flow cytometry | 0.23 ± 0.03 bits |
| RNA labeling (HCR) | Flow cytometry | 0.09 ± 0.03 bits |

* mean ± sample standard deviation of three replicates

The quality of an RNA measurement method is the result of a combined effect from multiple measurement steps. So, to assess how differences in single-cell RNA measurement quality might be related to specific steps of the measurement method, we compared the channel capacities of single-cell methods that differed only by one step in the measurement process (sample preparation or signal detection). First, the measurement quality is generally higher for RNA measurements that used microscopy for signal detection versus those that used flow cytometry. For example, microscopy measurements had a higher channel capacity than flow cytometry measurements for both FISH (≈ 1.06 bits vs. ≈ 0.23 bits; Tables 1 and 2) and HCR (≈ 0.88 bits vs. ≈ 0.09 bits). This is not surprising because microscopy allows for visual confirmation of cell-specific signal, and optimization of signal integration during image collection. Second, with both signal detection methods, we found that the measurement quality was better for FISH labeling versus HCR labeling. For example, FISH had a higher channel capacity than HCR, for both microscopy (≈ 1.06 bits vs. ≈ 0.88 bits) and flow cytometry (≈ 0.23 bits vs. ≈ 0.09 bits). This difference between RNA labeling methods could be attributed to the efficiency of probe hybridization to the target RNA, which was estimated to be higher for FISH than HCR in the experimental study [10].

To evaluate the quality of single-cell fluorescent protein measurements in bits, we compared the channel capacities of different methods of measuring the same fluorescent protein expression system. The different measurement methods included two commonly used antibiotic treatments to halt fluorescent protein translation prior to flow cytometry (kanamycin, chloramphenicol), as well as measurements of fluorescent protein in cells that had been labeled

Table 2. Detailed information for analysis and results from each method and replicate.

| Sample Preparation | Measurand | Signal Detection | $N_B$ | Channel capacity, Replicate 1 | Channel capacity, Replicate 2 | Channel capacity, Replicate 3 |
|---|---|---|---|---|---|---|
| Antibiotic treatment (Chloramphenicol, Cm) | Protein | Flow cytometry | 80 | 1.58 | 1.63 | 1.61 |
| RNA labeling (FISH) | Protein | Flow cytometry | 20 | 0.14 | 0.21 | 0.21 |
| RNA labeling (FISH) | RNA | Flow cytometry | 20 | 0.24 | 0.19 | 0.25 |
| RNA labeling (HCR) | Protein | Flow cytometry | 80 | 0.95 | 0.96 | 1.04 |
| RNA labeling (HCR) | RNA | Flow cytometry | 80 | 0.08 | 0.06 | 0.12 |
| Antibiotic treatment (Kanamycin, Kn) | Protein | Flow cytometry | 160 | 1.61 | 1.6 | 1.55 |
| RNA labeling (FISH) | Protein | Microscopy | 40 | 0.19 | 0.47 | 0.5 |
| RNA labeling (FISH) | RNA | Microscopy | 80 | 1.00 | 1.02 | 1.15 |
| RNA labeling (HCR) | Protein | Microscopy | 40 | 1.26 | 1.29 | 1.31 |
| RNA labeling (HCR) | RNA | Microscopy | 80 | 0.93 | 1.08 | 0.64 |

$N_B$ is the number of bins used to construct the empirical RNA or protein expression distributions (i.e., the probability transition matrix) for each measurement method. The resulting channel capacity results are also given for each biological replicate.

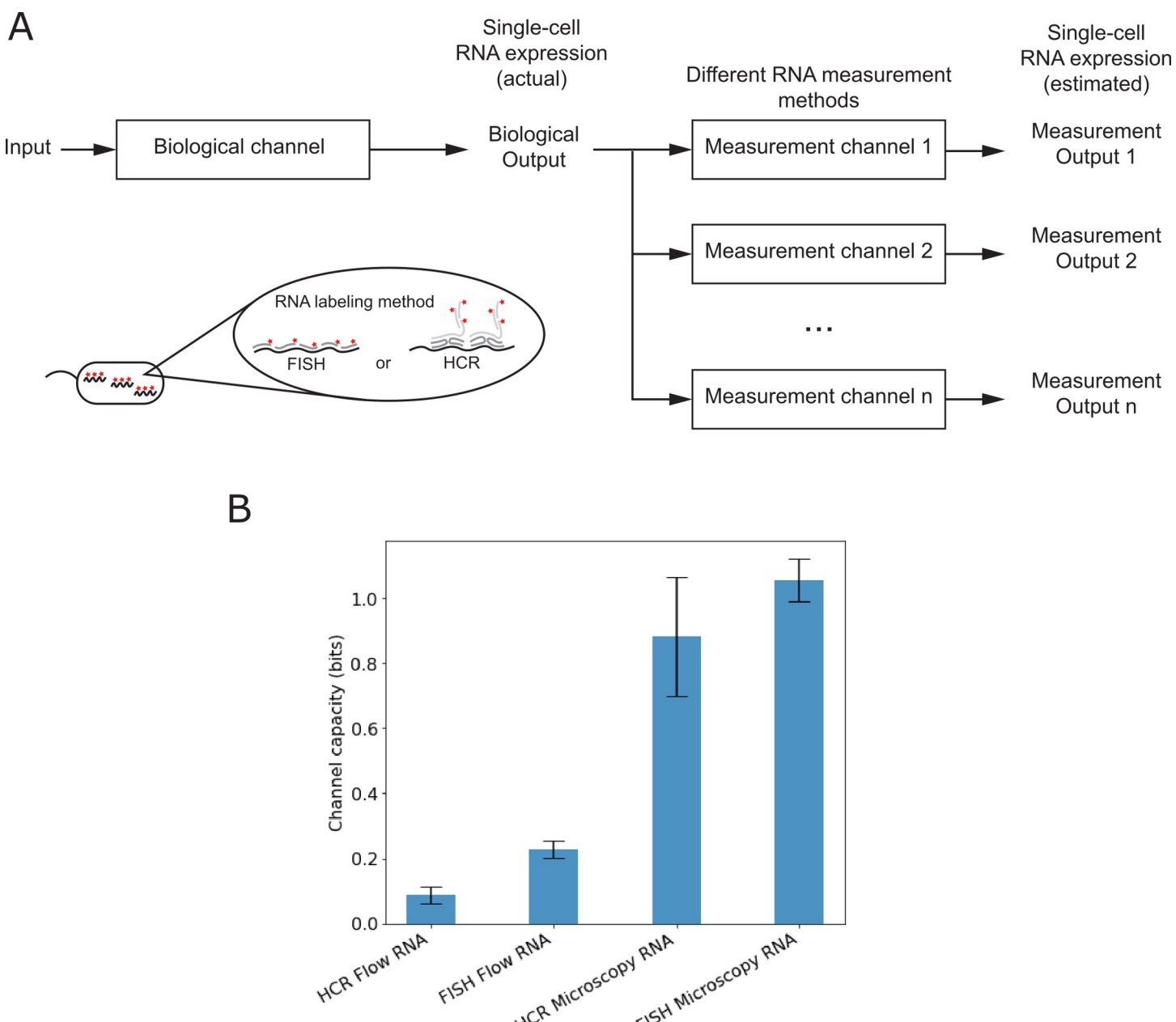

**Fig 2. Single-cell measurement quality of RNA expression, in bits.** (A) Different single-cell methods of measuring RNA use different steps for sample preparation and signal detection. Examples of sample preparation include RNA-labeling methods such as fluorescence *in situ* hybridization (FISH) or hybridization chain reaction (HCR) (inset). Examples of signal detection include microscopy and flow cytometry. When different single-cell methods are used to analyze the same biological output, the channel capacity between input and different measurement outputs can be used to compare single-cell measurement quality of RNA, in bits. (B) Channel capacities of different single-cell methods of estimating RNA from the same biological channel (mean +/- standard deviation of three biological replicates).

for RNA detection using FISH or HCR. For different fluorescent protein measurement methods, we observed a wide range of channel capacities (Fig 3, Tables 2 and 3). Flow cytometry detection of fluorescent protein following FISH labeling had the lowest channel capacity ($\approx$ 0.19 bits), while flow cytometry detection after antibiotic treatment had the highest channel capacity ($\approx$ 1.6 bits).

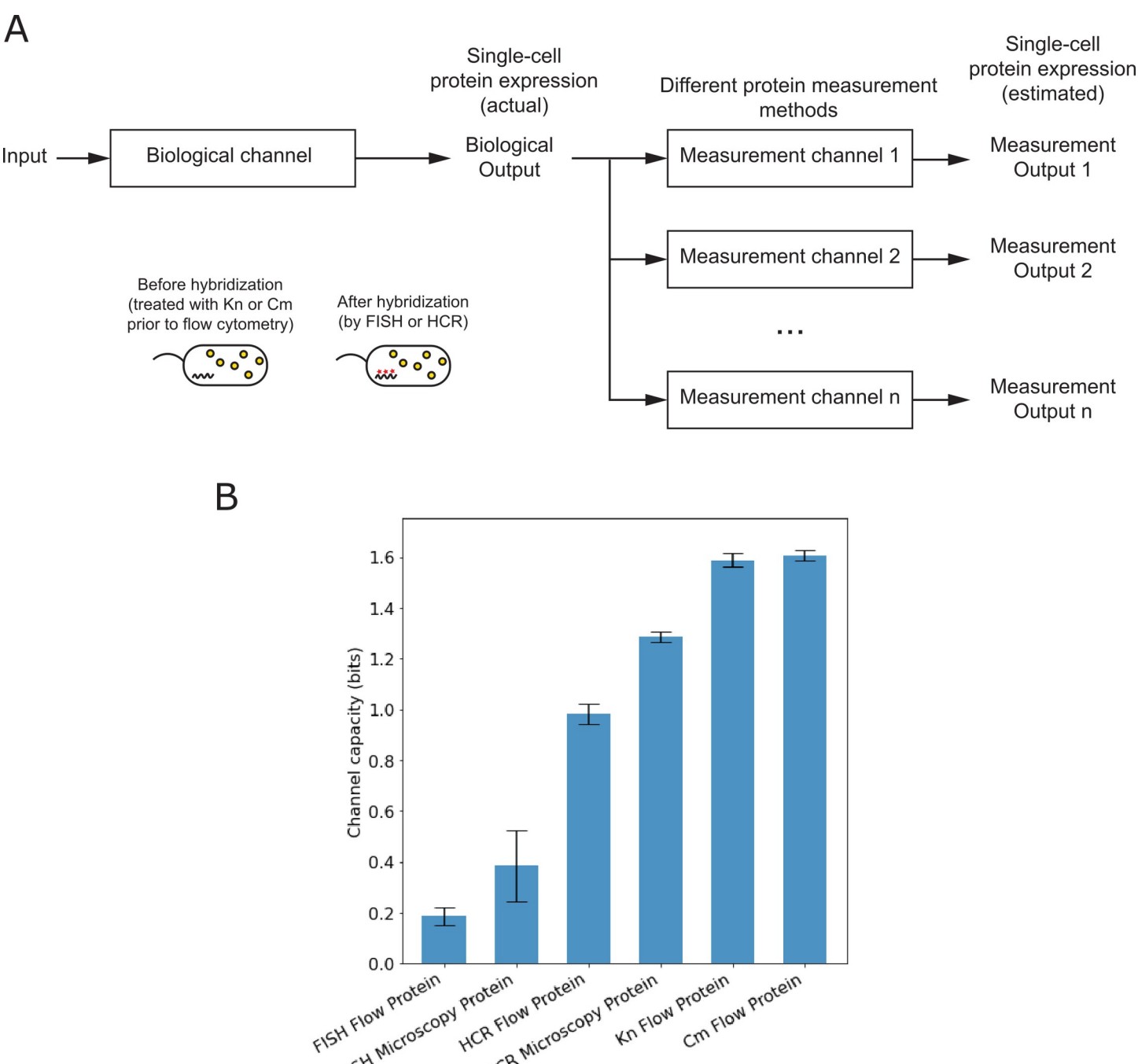

**Fig 3. Single-cell measurement quality of fluorescent protein expression, in bits.** (A) Different single-cell methods of measuring fluorescent protein use different steps for sample preparation and signal detection. Different antibiotic treatments (kanamycin, Kn; chloramphenicol, Cm) can be used to halt translation prior to fluorescent protein measurement by flow cytometry. Fluorescent protein can also be detected in cells following RNA-labeling methods such as fluorescence *in situ* hybridization (FISH) or hybridization chain reaction (HCR) (inset). Examples of signal detection include microscopy and flow cytometry. When different single-cell methods are used to analyze the same biological output, the channel capacity between input and different measurement outputs can be used to compare single-cell measurement quality of fluorescent protein expression, in bits. (B) Channel capacities of different single-cell methods of estimating fluorescent protein expression from the same biological channel (mean ± standard deviation of three biological replicates).

**Table 3. Channel capacities of single-cell fluorescent protein measurements.**

| Sample Preparation | Signal Detection | Channel capacity from environmental input to measurement output* |
|---|---|---|
| Antibiotic treatment (Chloramphenicol, Cm) | Flow cytometry | 1.61 ± 0.03 bits |
| Antibiotic treatment (Kanamycin, Kn) | Flow cytometry | 1.59 ± 0.03 bits |
| RNA labeling (HCR) | Microscopy | 1.29 ± 0.03 bits |
| RNA labeling (HCR) | Flow cytometry | 0.98 ± 0.05 bits |
| RNA labeling (FISH) | Microscopy | 0.39 ± 0.17 bits |
| RNA labeling (FISH) | Flow cytometry | 0.19 ± 0.04 bits |

* mean ± sample standard deviation of three replicates

Differences in measurement quality for different fluorescent protein methods result from specific steps within the measurement process. So, to assess how measurement quality might be related to specific steps of the measurement method, we compared the channel capacities of single-cell methods that differed by only one step in the measurement process (sample preparation or signal detection). First, as with RNA measurements, protein measurement quality was generally higher for microscopy than flow cytometry measurements. This held true for both FISH and HCR (Tables 2 and 3). Second, protein measurement quality decreased when cells were labeled for RNA detection. This can be seen by comparing the flow cytometry results for fluorescent protein measurements after the two antibiotic treatments (channel capacity ≈ 1.6 bits) to those made after FISH or HCR labeling (≤ 1.0 bits). Finally, unlike RNA measurements, protein measurement quality was generally higher after HCR labeling than after FISH labeling, regardless of the signal detection method. This could be due to different effects that the RNA-labeling buffers have on fluorescent protein signal within the cell.

## Discussion

Previous studies have estimated information transmission between an environmental stimulus and single-cell measurements of gene expression [20–22]. However, the role of single-cell measurement quality is largely ignored in evaluation of these biological processes, and information loss through the measurement process is not directly estimated. We have shown how to assess and compare the measurement quality of different single-cell methods using the channel capacity between an environmental stimulus and the measured response. This provides a practical and intuitive way to compare information loss due to different single-cell measurement methods. The approach described here is generalizable to assess and compare the measurement quality of other single-cell methods, including different data analysis methods. For more complex, multi-variate single-cell measurements (e.g., multi-transcript RNA-seq, time-series microscopy), application of our approach to compare different measurement protocols and/or data analysis methods would probably require an alternative to the Blahut-Arimoto algorithm for estimating the channel capacity [23]. Our general approach should still be valid for those types of data, however: the highest quality method will be the one that results in the highest channel capacity. Hence, we anticipate that this approach will increase the adoption of information theory as a practical and universal way to assess the quality of single-cell measurements. Finally, we note that any channel capacity estimate using finite data represents a lower bound on information transmission [22]. So, by estimating channel capacity from environmental input through single-cell measurements, we provide a lower bound on the channel capacity for both the biological system and the measurement system. The approach we demonstrate here shows how the analysis of information transmission through measurement

processes enables universal comparability not only between different measurements of biology, but also between measurements and biology itself.

## Materials and methods

### Source of experimental data

Single-cell measurement data was analyzed from a recently-reported study [10]. Briefly, experimental measurements were performed as follows: *E. coli* cells were grown in cultures containing different concentrations of IPTG, which served as an environmental stimulus that induced expression of *eyfp* RNA and eYFP protein. Each culture was divided (split) for different sample preparations, including different antibiotic treatments (kanamycin or chloramphenicol), or different RNA labeling strategies (FISH [2, 24–29] or HCR [30]). Following sample preparation, RNA and fluorescent protein expression were measured using two different signal detection methods: microscopy and flow cytometry. With FISH and HCR microscopy, single-molecule localization was used to estimate the distribution of the *eyfp* RNA copy number per cell using well-established techniques [24, 31]. With flow cytometry, the distribution of the fluorescence signal per cell was determined using an automated gating algorithm [32]. In this manner, multiple single-cell measurement methods were performed in parallel with minimal and well-defined experimental variability. The results of all single-cell measurements are publicly-available through the NIST Data Portal (https://doi.org/10.18434/mds2-2300).

### Computation of channel capacity

The channel capacity for each measurement method was computed numerically using the Blahut-Arimoto algorithm (Fig 4) [18, 19].

   **Binning and discretization of single-cell measurements.**   The Blahut-Arimoto algorithm requires discrete distributions at each input signal level. To apply the algorithm to single-cell data, continuous measurement results (e.g. a list of real numbers) were discretized by binning the data with equal-width bins spanning the range of measurement results. The resulting discrete probability distributions were used directly as the discrete transition probability matrices as detailed below.

   As described in previous publications, the choice of bin size can affect the calculated channel capacity, and the process of choosing the optimal number of bins is heuristic [33, 34]. If the number of bins is too low, the mutual information and channel capacity are underestimated. But, if the number of bins is too high, the mutual information and channel capacity are overestimated. Typically, a range of bin numbers can be found over which the channel capacity does not depend sensitively on the number of bins used. So, in this work, the number of histogram bins was chosen based on comparisons of channel capacity values obtained for different numbers of bins, according to the following procedure:

   Equal-width histogram bins were used, with the minimum and maximum bins set to span the full range of the observations for each dataset. For each transcript or protein expression dataset, the Freedman-Diaconis' rule was used to calculate a recommended bin width. Then, for each measurement method, an initial bin width was chosen as approximately ten times the mean recommended bin width over the three replicates of the method. The channel capacity was computed using the resulting transition probability matrix. Then, the bin width was decreased by a factor of 2 (i.e., number of bins increased 2-fold), and the channel capacity was computed again. For each measurement method, if the mean channel capacity increased by more than 0.1 bits, the bin with was decreased again by a factor of 2 and the channel capacity re-calculated. When the resulting change in the mean channel capacity was less than 0.1 bits, the channel capacity values from the previous bin width were used. Fig 5 shows the results for

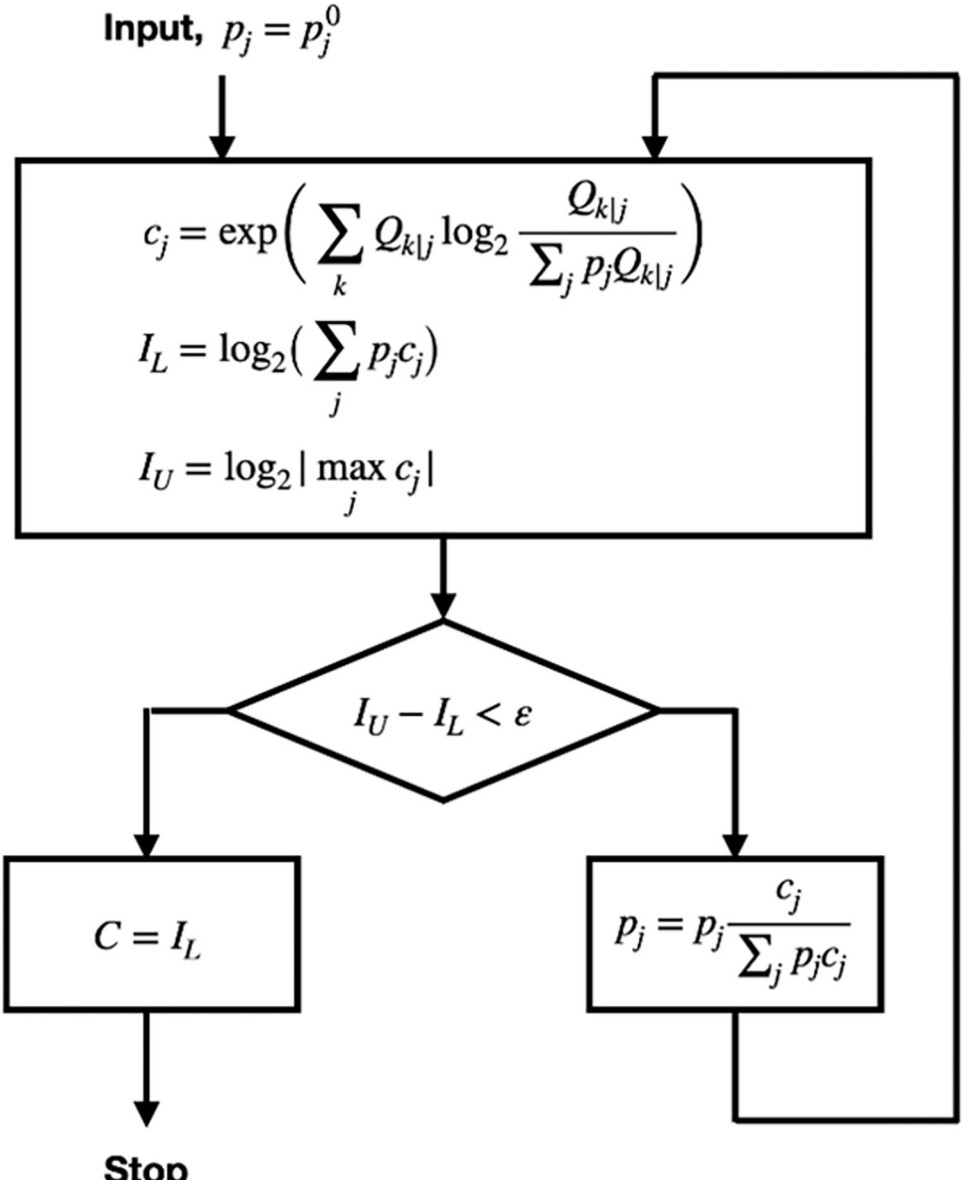

**Fig 4. Flowchart of the Blahut-Arimoto algorithm to compute channel capacity (adapted from Blahut, 1972 [18]).**

the channel capacity calculated using different numbers of bins, and Table 3 lists the number of bins used for each measurement method.

### Implementation of the Blahut-Arimoto algorithm

Here, we briefly describe the Blahut-Arimoto algorithm using the same notation as used in Blahut's 1972 paper [18]. Mutual information through an information channel is

$$I(p, Q) = \sum_j \sum_k p_j Q_{k|j} \log_2 \frac{Q_{k|j}}{\sum_j p_j Q_{k|j}} \tag{1}$$

where $Q$ is the probability transition matrix, constructed in our case from the discretized

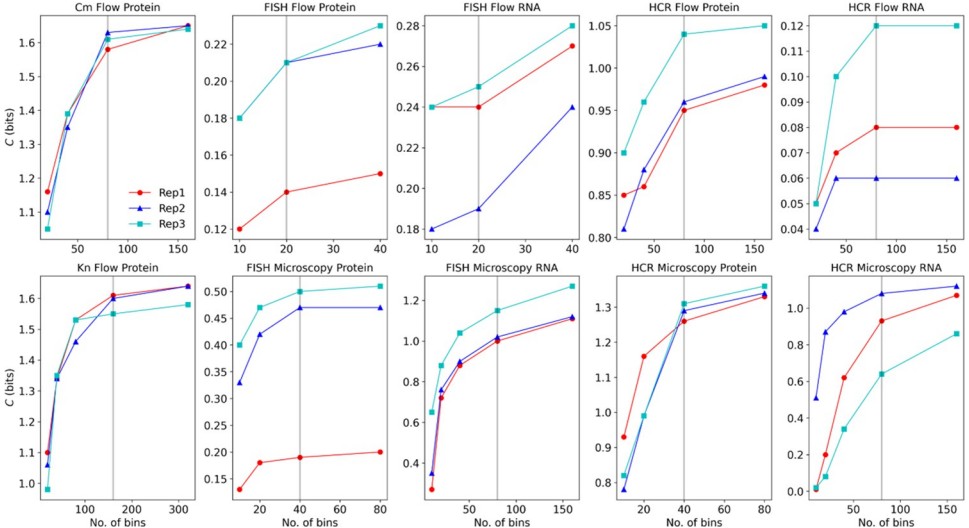

**Fig 5. Dependence of channel capacity on the number of bins used to construct the empirical RNA or protein distributions for each of the methods.** The vertical gray line shows the number of bins used to compute the channel capacity reported in the manuscript.

empirical RNA or protein distributions, and $p$ is the discrete input probability distribution. The channel capacity is the maximum mutual information over all possible input distributions

$$C = \max_p I(p, Q) \tag{2}$$

The Blahut-Arimoto algorithm solves the maximization problem, Eq (2), using an additional property of the mutual information function,

$$I(p, Q) = \max_P J(p, Q, P) = \max_P \sum_j \sum_k p_j Q_{k|j} \log_2 \frac{P_{j|k}}{p_j} \tag{3}$$

where $P$ is a variable transition matrix from the output variable to the input variable. Combining Eqs (2) and (3) we obtain

$$C = \max_p \ \max_P J(p, Q, P). \tag{4}$$

The Blahut-Arimoto algorithm is based on the idea that for a fixed input distribution $p$ the transition matrix $P$ that maximizes $J(p, Q, P)$ is

$$P_{j|k}^* = \frac{p_j Q_{k|j}}{\sum_j p_j Q_{k|j}} \tag{5}$$

and for a fixed output-to-input transition matrix, $P$, the input distribution that maximizes $J(p, Q, P)$ is

$$p_j = \frac{\exp(\sum_k Q_{k|j} \log_2 P_{j|k})}{\sum_j \exp(\sum_k Q_{k|j} \log_2 P_{j|k})} \tag{6}$$

The Blahut-Arimoto algorithm is an iterative method to estimate the channel capacity and the optimal input distribution (i.e., the distribution, $p_j$, that maximizes the mutual information to give the channel capacity). The algorithm is initialized with a starting guess for the input

distribution, $p_j^0$. As shown in the original papers by Blahut and Arimoto [18, 19], the algorithm is guaranteed to monotonically approach the exact result for the channel capacity. So, if enough iterations are run, the resulting channel capacity estimates will not depend sensitively on the starting guess, $p_j^0$. For simplicity, we used a uniform discrete distribution, i.e., $p_j^0 = 1/N_{input}$ for each $j$, where $N_{input}$ is the number of input levels measured ($N_{input} = 8$ in the current work, so $p_j^0 = 0.125$). At each iteration of the algorithm, Eqs (5) and (6) are used to get an updated estimate for the optimal input distribution and the channel capacity. The algorithm is stopped when the change between iterations is smaller than a predefined value, $\varepsilon$, which, for this work was set to $10^{-4}$. Since this value is much smaller than the typical uncertainty (see Tables 1–3), the results won't depend sensitively on either the starting guess, $p_j^0$, or the value of the stopping criterion, $\varepsilon$.

The flowchart for the final algorithm is shown in Fig 4, where the key variables are defined as follows:

$Q_{k|j}$—Probability transition matrix from the input to the output, which numerically defines the information channel. This matrix is the main input to the Blahut-Arimoto algorithm. It is the set of conditional distributions of the output for each fixed values of the input. The empirical distribution of RNA or protein expression for each input level (IPTG concentration) is the $j^{th}$ column of $Q_{k|j} : Q_{k|j} = \frac{n_{k|j}}{N_j}$, where $n_{k|j}$ is the number of cells from the $j^{th}$ sample with a transcript or protein measurement falling in the $k^{th}$ discretization bin, and $N_j$ is the total number of cells from the $j^{th}$ sample. The procedure for choosing the number of histogram bins is described above.

$p_j^0$—Initial guess for the optimal input distribution that achieves channel capacity. This vector has the same dimension as the number of input concentrations.

$\varepsilon$—Numerical threshold value for the stopping condition of the iterative algorithm.

## Acknowledgments

The authors thank Greta Babakhanova, Joseph Hubbard, Jacob Majikes, Anne Plant, and Elizabeth Strychalski for useful discussions and critical reviews of the manuscript.

**Disclaimer:** The National Institute of Standards and Technology (NIST) notes that certain commercial equipment, instruments, and materials are identified in this paper to specify an experimental procedure as completely as possible. In no case does the identification of particular equipment or materials imply a recommendation or endorsement by NIST, nor does it imply that the materials, instruments, or equipment are necessarily the best available for the purpose.

## Author Contributions

**Conceptualization:** Jayan Rammohan, David Ross.

**Formal analysis:** Swarnavo Sarkar.

**Writing – original draft:** Jayan Rammohan, Swarnavo Sarkar, David Ross.

**Writing – review & editing:** Jayan Rammohan, Swarnavo Sarkar, David Ross.

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
