## [Decision Letter · Decision Letter 0]

18 Jan 2022

PONE-D-21-36062Single-cell measurement quality in bitsPLOS ONE

Dear Dr. Ross,

Thank you for submitting your manuscript to PLOS ONE. After careful consideration, we feel that it has merit but does not fully meet PLOS ONE’s publication criteria as it currently stands. Therefore, we invite you to submit a revised version of the manuscript that addresses the points raised during the review process.

Please pay attention to the comments of all reviewers and in particular reviewer #3.  You need to address all their comments in the revision.

We look forward to receiving your revised manuscript.

Kind regards,

Panayiotis V. Benos, PhD

Academic Editor

PLOS ONE

Journal Requirements:

Reviewers' comments:

Reviewer's Responses to Questions

**Comments to the Author**

1. Is the manuscript technically sound, and do the data support the conclusions?

Reviewer #1: Yes

Reviewer #2: Yes

Reviewer #3: Yes

2. Has the statistical analysis been performed appropriately and rigorously? 

Reviewer #1: Yes

Reviewer #2: Yes

Reviewer #3: No

3. Have the authors made all data underlying the findings in their manuscript fully available?

Reviewer #1: Yes

Reviewer #2: Yes

Reviewer #3: Yes

4. Is the manuscript presented in an intelligible fashion and written in standard English?

Reviewer #1: Yes

Reviewer #2: Yes

Reviewer #3: Yes

5. Review Comments to the Author

Reviewer #1: General comments:

The manuscript is well written and understandable and conclusions in general appear well supported by the data presented in the manuscript. Overall I have two general points that could be added to the manuscript:

1) Data analyzed in this manuscript was published beforehand by the same laboratory (PMID:34079048), as the authors state. However, it would be helpful to summarize how the current analysis and its conclusions relate to what has been published before.

2) It appears that if different environmental inputs/stimuli would be used, channel capacity estimates could change. Is the expected in practice? Would it have the potential to affect observations reported, for example that for RNA microscopy has typically higher capacity than flow cytometry? If such a caveat exists, it would be good to point this out in the manuscript.

Specific comments

There are instances where capacity reported in the text is different from what is in the corresponding table. For instance, on page 11 the text quotes the capacity for HCR labeled RNA as 0.08 bits, whereas table 1 shows 0.09 bits; in the text the capacity of FISH-labeled RNA and microscopy is quoted as 0.97 bits, whereas table 1 shows 1.06 +/- 0.07 bits, which puts the value in the text outside of three standard deviations from what is in the table. It would be helpful, throughout the manuscript, if capacities in the text would match information given in tables.

Reviewer #2: The authors use mutual information measure to compare different experimental methods of single cell expression analysis. It is a well-controlled analysis with a small number of variables. it isn't exactly clear how useful it will be for large scale experiments, such as scRNA-seq, but they state that it should be generalizable. Perhaps they will do that in future work. I do not find any technical problems with the work or the presentation so I conclude that it meets the PLoS One criteria for publication.

Reviewer #3: In this article, the authors proposed an information theory approach for

assessing and compare single-cell measurement quality in bits. They then claim that it provides a

universally comparable metric for information content.

1- Although the article tackle a very important issue, the manuscript is not detailed enough.

- More information and detail explanations between single-cell measurement and information theory should be given.

- The methodology section on the computation of the channel capacity is about 5 lines with references to supporting materials where more details are given. I think these details should be part of the manuscript itself to allow the readers to have a flow.

2- In Table 1 and Table 2, what is the ground true?

3- In the support materials, page 2 the authors said: "This vector has the same dimension as the number of input concentrations, we chose an uniform discrete distribution in our work, i.e., p_j^0=0.125 for each j. ε - Numerical error value for convergence, we chose ε=〖10〗^(-4)."

- More explanations should be given. Why a normal distribution, why 0.125 and 10^-4, how do the results vary relative to these parameters. A sensitivity analysis of these parameters should be performed.

4 - There are some texts similarities between this manuscript and the manuscript [reference 10] recently published by the same authors. Some of these parts should be rewritten.

6. PLOS authors have the option to publish the peer review history of their article (what does this mean?). If published, this will include your full peer review and any attached files.

Reviewer #1: No

Reviewer #2: No

Reviewer #3: No

---

## [Author Response · Author response to Decision Letter 0]

2 Mar 2022

We thank the editors and reviewers for the thoughtful reviews of our manuscript. We have made changes to the manuscript in response to each of the reviewers’ comments as detailed below. In general, we have tried to clarify the descriptions of our methods, and we have moved the content that was in the SI (detailed descriptions of the algorithms used) to the Materials and Methods section of the manuscript.

Reviewer #1: General comments:

The manuscript is well written and understandable and conclusions in general appear well supported by the data presented in the manuscript. Overall I have two general points that could be added to the manuscript:

1) Data analyzed in this manuscript was published beforehand by the same laboratory (PMID:34079048), as the authors state. However, it would be helpful to summarize how the current analysis and its conclusions relate to what has been published before.

Response: In the second sentence of the introduction, we added text to more clearly summarize the relationship between the previous publication and the current manuscript: “… we used a recently-reported collection of data from different single-cell measurements… [10]. In that study, a split-sample approach was used to measure cells harvested from the same replicate cultures… Here, we use the experimental results from [10] to show how channel capacity can be used as a metric…”

2) It appears that if different environmental inputs/stimuli would be used, channel capacity estimates could change. Is the expected in practice? Would it have the potential to affect observations reported, for example that for RNA microscopy has typically higher capacity than flow cytometry? If such a caveat exists, it would be good to point this out in the manuscript.

Response: Yes, different inputs/stimuli would result in different channel capacity estimates. 

We have added a new paragraph (page 5 of the revised manuscript) to make this point clear:

“As a final consideration, it is important to note that the channel capacity can also depend on the choice of values used for the input stimuli. If the channel capacity is the logarithm of the number of distinguishable input levels, then it clearly cannot be greater than the logarithm of the number input levels measured. For example, if an experiment only uses two input levels (e.g., test and control, or high and low), then the channel capacity determined by our approach will always be less than or equal to one (log2(2) = 1). Furthermore, to obtain the best estimates of the biological channel capacity and the best comparison of different methods, the input levels should be chosen to span the full range of biological response. For example, the datasets used here include IPTG concentrations across the full induction curve, with input levels that result in low biological output (i.e., gene expression), high biological output, and intermediate biological output. In general, a different choice of input levels measured in an experiment could lead to different estimates of the channel capacity with our approach. So, for assessment of measurement quality, we recommend comparing only methods implemented with the same set of input levels (as with the split-sample dataset used here).”

Specific comments

There are instances where capacity reported in the text is different from what is in the corresponding table. For instance, on page 11 the text quotes the capacity for HCR labeled RNA as 0.08 bits, whereas table 1 shows 0.09 bits; in the text the capacity of FISH-labeled RNA and microscopy is quoted as 0.97 bits, whereas table 1 shows 1.06 +/- 0.07 bits, which puts the value in the text outside of three standard deviations from what is in the table. It would be helpful, throughout the manuscript, if capacities in the text would match information given in tables.

Response: We thank the reviewer for pointing out these discrepancies. We have carefully compared the text and the tables to ensure the values match in the revised manuscript. We also re-checked the standard deviations used in the tables and found that we had mistakenly used the population standard deviation. So, we re-calculated the standard deviations reported in the table (using the sample standard deviation). 

Reviewer #2: The authors use mutual information measure to compare different experimental methods of single cell expression analysis. It is a well-controlled analysis with a small number of variables. it isn't exactly clear how useful it will be for large scale experiments, such as scRNA-seq, but they state that it should be generalizable. Perhaps they will do that in future work. I do not find any technical problems with the work or the presentation so I conclude that it meets the PLoS One criteria for publication.

Response: We have added an additional sentence to the Discussion to briefly address the generalizability to measurements such as scRNA-seq (page 13 of the revised manuscript):

“For more complex, multi-variate single-cell measurements (e.g., multi-transcript RNA-seq, time-series microscopy), application of our approach to compare different measurement protocols and/or data analysis methods would probably require an alternative to the Blahut-Arimoto algorithm for estimating the channel capacity [23]. Our general approach should still be valid for those types of data, however: the highest quality method will be the one that results in the highest channel capacity.”

Reviewer #3: In this article, the authors proposed an information theory approach for

assessing and compare single-cell measurement quality in bits. They then claim that it provides a

universally comparable metric for information content.

1- Although the article tackle a very important issue, the manuscript is not detailed enough.

- More information and detail explanations between single-cell measurement and information theory should be given.

- The methodology section on the computation of the channel capacity is about 5 lines with references to supporting materials where more details are given. I think these details should be part of the manuscript itself to allow the readers to have a flow.

Response: We agree with the reviewer and have moved the description of the Blahut-Arimoto algorithm and our specific implementation to the Materials and Methods section of the manuscript. We have also edited that text and the brief description of the approach in the Results section to more clearly describe the methods used.

2- In Table 1 and Table 2, what is the ground true?

Response: The closest thing to “ground truth” in this case is the true biological channel capacity, which will be greater than the measured channel capacities. So, the best estimate of the biological channel capacity is the highest measured channel capacity (corresponding to the highest quality measurement method).

We have edited the text (page 7 of the revised manuscript) to clarify this point:

“Since no measurement is perfect, the measurement channel will degrade the information that it transmits. So, the measured channel capacity, i.e. the channel capacity between the input and the estimated gene expression, will always be less than the biological channel capacity, i.e., the channel capacity between the input and the actual gene expression. Higher quality measurements, however, will degrade the information less. So, we can assess relative measurement quality by comparing the measured channel capacities for different measurement methods: Higher quality measurements will result in a higher measured channel capacity. (i.e., closer to the true biological channel capacity).”

3- In the support materials, page 2 the authors said: "This vector has the same dimension as the number of input concentrations, we chose an uniform discrete distribution in our work, i.e., p_j^0=0.125 for each j. ε - Numerical error value for convergence, we chose ε=〖10〗^(-4)."

- More explanations should be given. Why a normal distribution, why 0.125 and 10^-4, how do the results vary relative to these parameters. A sensitivity analysis of these parameters should be performed.

Response: We have moved the relevant sections from the SI to the Methods section and revised them to improve the clarity. In particular, in the revised manuscript, we point out that the Blahut-Arimoto algorithm converges monotonically toward exact channel capacity value (as shown in the 1972 papers by Blahut and Arimoto), so the result does not depend on the choice of p_j^0. Furthermore, we explain that the parameter ε determines how close the iterative algorithm must get to the exact answer before the iterations are stopped. So, we chose ε (0.0001 bits) to be much small than the typical uncertainty in the channel capacity estimates (>= 0.03 bits, from measurement replicates, see manuscript Tables 1-3).

4 - There are some texts similarities between this manuscript and the manuscript [reference 10] recently published by the same authors. Some of these parts should be rewritten.

Response: This point is difficult to address without more specific information from the reviewer (i.e., which parts of the manuscript have text too similar to the previous publication). However, we carefully compared the text from the two manuscripts, paying particular attention to paragraphs we thought might have overlap. Although the two manuscripts have some overlap in language (they are related manuscripts), we only found one instance of a string longer than four words that was identically repeated in both manuscripts; “… in parallel on cells harvested from the same original culture.” In the revised manuscript, we have modified the text to avoid using the same sentence fragment as in the previous publication.

---

## [Editor Report · Decision Letter 1]

18 May 2022

Single-cell measurement quality in bits

PONE-D-21-36062R1

Dear Dr. Ross,

We’re pleased to inform you that your manuscript has been judged scientifically suitable for publication and will be formally accepted for publication once it meets all outstanding technical requirements.

Kind regards,

Panayiotis V. Benos, PhD

Academic Editor

PLOS ONE
---

## [Editor Report · Acceptance letter]

2 Aug 2022

PONE-D-21-36062R1 

Single-cell measurement quality in bits 

Dear Dr. Ross:

I'm pleased to inform you that your manuscript has been deemed suitable for publication in PLOS ONE. Congratulations! Your manuscript is now with our production department. 

Kind regards, 

on behalf of

Professor Panayiotis V. Benos 

Academic Editor

PLOS ONE